# Variations in Structure among Androecia and Floral Nectaries in the Inverted Repeat-Lacking Clade (*Leguminosae: Papilionoideae*)

**DOI:** 10.3390/plants11050649

**Published:** 2022-02-27

**Authors:** Andrey Sinjushin, Maria Ploshinskaya, Ali Asghar Maassoumi, Mohammad Mahmoodi, Ali Bagheri

**Affiliations:** 1Department of Genetics, Faculty of Biology, Lomonosov Moscow State University, Leninskie Gory 1-12, 119234 Moscow, Russia; 2Department of Higher Plants, Faculty of Biology, Lomonosov Moscow State University, Leninskie Gory 1-12, 119234 Moscow, Russia; ploshinskaya@rambler.ru; 3Botany Research Division, Research Institute of Forests and Rangelands, Agricultural Research, Education and Extension Organization (AREEO), Tehran 13185-116, Iran; maassoumi@rifr-ac.ir (A.A.M.); mahmoodi@rifr-ac.ir (M.M.); 4Department of Plant and Animal Biology, Faculty of Biological Science and Technology, University of Isfahan, Isfahan 81746-73441, Iran; a.bagheri@sci.ui.ac.ir

**Keywords:** diadelphy, monadelphy, pollination, secretion, stamen

## Abstract

The vast majority of highly valuable species of the Leguminosae in temperate latitudes belong to the Inverted Repeat-Lacking Clade (IRLC). Despite having a generally conserved monosymmetric floral morphology, members of this group are remarkable with a pronounced diversity of floral sizes, modes of staminal fusion, and pollination strategies. This paper examined androecia and floral nectaries (FNs) in selected genera of the IRLC. External morphology was investigated using stereomicroscopy and scanning electron microscopy. In some cases, the pattern of staminal fusion was additionally examined in transverse sections using light microscopy. Androecia of all selected genera fell into one of four types, viz., monadelphous, pseudomonadelphous, diadelphous or diadelphous reduced (with inner stamens converted into sterile staminodes). However, there was significant variation in the stamens’ mode of contact, as well as the shape and size of the fenestrae providing access to FNs. Some types seemed to arise independently in different genera, thus providing a high level of homoplasy. FNs were more conserved and comprised areas of secretory stomata in the abaxial part of the receptacle and/or hypanthium. Nectariferous stomata could be found in very miniaturized flowers (*Medicago lupulina*) and could even accompany monadelphy (*Galega*). This indicates that preferential self-pollination may nevertheless require visitation by insects.

## 1. Introduction

The pronounced evolutionary success of angiosperms is connected with the outstanding variety in their reproductive strategies, mostly conditioned by the structural and functional diversity of their flowers. These display a broad range of adaptations towards different modes of pollination, using either their own (autogamy) or other plants’ (allogamy) pollen. The third largest angiosperm family, Leguminosae, is remarkable with its flowers being highly diverse in size, symmetry and merism, as well as its interplay between different parts. In addition to the fundamental insight into floral evolution in this family and angiosperms as a whole, studies on floral morphology and evolution in Leguminosae are of significant practical interest, as this group holds the world record with respect to number of cultivated species [1]. Numerous leguminous species from temperate latitudes, such as pea (*Pisum sativum* L.), faba bean (*Vicia faba* L.) and other cultivated vetches, grass pea (*Lathyrus sativus* L.) and other valuable species of the same genus, lentil (*Lens culinaris* Medik.), chickpea (*Cicer arietinum* L.), and many others, belong to the Inverted Repeat-Lacking Clade (IRLC) of the subfamily Papilionoideae [2]. This group also includes genera of high ornamental, forage, melliferous or medicinal value, such as *Trifolium*, *Melilotus*, *Medicago*, *Glycyrrhiza*, *Wisteria,* etc.

A flower groundplan is conserved across the IRLC: representatives of this clade are characterized by pentamerous pentacyclic flowers and differentiation of corolla into three petal types, i.e., the so-called papilionate corolla or ‘flag blossom’ [3]. In spite of similar gross morphology, flowers from different lineages of the IRLC differ mostly with respect to the structure of the androecium, especially in terms of synorganization between stamens [4]. Stamens may be completely free, but more often fuse in some way or other. In many genera, the adaxial inner (vexillary) stamen remains free, whereas the other nine stamens fuse, producing an incomplete tube, which is referred to as a diadelphous androecium. In some cases, all ten stamens unite in a fully closed tube (monadelphous androecium). A pseudomonadelphous androecium is formed when the vexillary stamen is postgenitally fused (or, more broadly, ‘secondarily reconnected’ as denoted by [5]; for discussion of the ambiguity of the term ‘fusion’, see below) with adjacent ones. In this case, one (rarely) or two holes (fenestrae or nectar windows) remain at the base of the vexillary stamen [6]. All four types of androecium are found in the IRLC, sometimes coexisting in one genus. For example, most members of the world’s largest angiosperm ‘megagenus’, *Astragalus*, possess diadelphous androecia [7]. However, monadelphous androecia have been reported in several species, such as *A. monadelphus* Maxim., *A. neomonadelphus* H.T.Tsai & T.F.Yu, and *A. donianus* DC. Careful anatomical investigation proved the pseudomonadelphous nature of the androecium in one of them, *A. monadelphus*, as the vexillary stamen, although adnate to adjacent filaments, retains its epidermis [8]. In at least two species of the same genus, *A. epiglottis* L. and *A. pelecinus* (L.) Barneby, stamens of the inner whorl become antherless, i.e., are converted into sterile staminodes [4].

The available descriptions of staminal fusion and, to an even greater degree, FNs in the papilionoid Leguminosae are often contradictory. For example, there was a long-lasting discussion regarding the androecial type in *Anthyllis* (tribe Loteae), finalized with an anatomical survey, which demonstrated that the vexillary filament, although tightly attached to adjacent ones, has its own epidermis, and hence the whole androecium cannot be classified as monadelphous [9]. The question of whether two floral structures are fused or simply in contact is not trivial [10]. The presence of separate epidermal layers of two adjacent organs can be interpreted as a sign that these organs are not subject to congenital fusion. However, in particular cases, the contacting epidermal cells can dedifferentiate [11], resulting in a situation histologically indistinguishable from true congenital fusion. Taking into account all these difficulties, Sokoloff et al. [12] discriminated between perfect (when the original epidermal layers dedifferentiate) and imperfect (when separate epidermal tissues persist) types of postgenital fusion. Authors of the cited paper [12] (p. 18) admitted a difficulty ‘to draw a clear boundary between imperfect postgenital fusion of the units and mere appression of their free surfaces’ and suggested the recognition of postgenital fusion in cases ‘when organs or their parts join each other and remain united by the end of all developmental processes’. Available reviews of papilionoid androecia refer to whether the initially free vexillary stamen filament ‘attaches itself’ [4] (is subject to ‘edge-to-edge fusion’: [3]; ‘reconnects’: [5]) to the adjacent filaments or remains free, without going deeper into the mechanism of this attachment. That is why, in order to look further forward, we interpreted (pseudo)monadelphy more broadly.

The ultimate goal of androecial fusion in a leguminous flower is to establish interaction with a pollinator (usually an insect, for the IRLC), especially to provide access to floral nectaries (FNs). As Rodríguez-Riaño et al. [4] concluded, monadelphous and pseudomonadelphous (lacking basal fenestrae) androecia are usually found in nectarless flowers.

The situation regarding FNs in the Leguminosae is also debatable in some cases, as there are several sources of information on FNs. These are (1) observations of nectar secretion and production, (2) examination of floral anatomy with a light microscope, and (3) targeted investigations of FNs using scanning and/or transmission electron microscopy. Only the latter approach unmistakably indicates if FNs are present in the flower and what their morphology is. As a result, contradictory data on FNs in legumes are available. For example, Gulyás and Kincsek [13] (p. 57) described flowers of *V. faba*, *Medicago sativa* L. and *P. sativum* as possessing FNs of the ‘epimorphic’ type, i.e., ‘located around the base of the gynoecium in the inner side of the receptacle’. These descriptions found no support in several investigations that, via SEM, indicated an abaxial position of the nectar-producing stomata [14,15,16]. Similarly, several papers reported that flowers of *Galega* produced no nectar [4,13,17], whereas others listed this genus among valuable melliferous plants [18] and described its diurnal dynamics of nectar secretion [19].

All these data, taken together, indicate that androecia and FNs form a complex that is decisive for the reproductive strategy of a species. However, morphological and evolutionary interactions between different androecium/FN types still await clarification. This work aims to investigate androecium and FN morphology in selected members of the IRLC with special reference to their variability and possible associations between them.

## 2. Results

### 2.1. Androecial Morphology

Following and expanding the terminology used by Rodríguez-Riaño et al. [4], androecia of examined plants could be classified into several types. As discussed in the Introduction, only those cases where the vexillary stamen is completely free from the adjacent ones are referred to as diadelphous.

Diadelphous: Caragana arborescens (Figure 1A), Astragalus albispinus (normal), A. caspicus, A. cicer, Colutea arborescens, Oxytropis kamtschatica, and most probably Wisteria sinensis (Appendix A). In A. cicer, Ca. arborescens, and O. kamtschatica, wide gaps remain on either side of the vexillary stamen, whereas in Co. arborescens two large fenestrae are formed at the base of this stamen.Monadelphous, with all ten stamens completely fused in an adaxially closed tube: A. albispinus (abnormal: Figure 1H), Ononis spinosa (Appendix A).

**Figure 1 plants-11-00649-f001:**
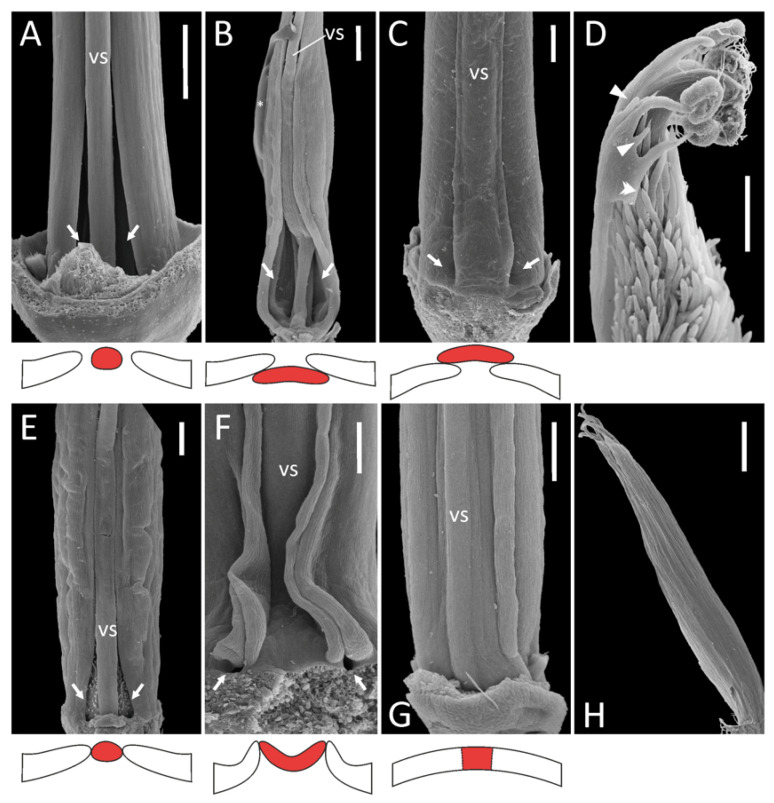
Androecia in selected representatives of the IRLC: overall morphology as seen from the adaxial side (above, SEM images) and schematic representation of the position of the vexillary stamen (below, marked with red). On all photos, the receptacle is oriented downwards and the perianth removed. (**A**), *Caragana arborescens*; (**B**), *Melilotus officinalis*; (**C**), *Lathyrus vernus*; (**D**), *Astragalus epiglottis*; (**E**), *Onobrychis viciifolia*; (**F**), *L. japonicus* subsp. *maritimus*; (**G**), *Galega officinalis*; (**H**), *A. albispinus* (abnormal specimen). Key: vs = vexillary stamen; arrows = fenestrae; arrowheads = staminodes; double arrowhead = unfused margin of staminal tube; asterisk = area covered with glue in the course of specimen mounting (**B**). Scale bars: 1 mm (**A**,**H**), 300 μm (**B**–**G**).

A detailed examination of an external surface of a staminal tube in both species of *Galega* revealed that the vexillary filament seems tightly fused with adjacent filaments, but its borders are still traceable (Figure 1G). Therefore, we examined the anatomy of the androecium in both species of *Galega*. This revealed that for *G. orientalis*, the vexillary filament is fused with the adjacent stamens and has a common epidermis with them (Figure 2A,B; see also Appendix A) but is the first to detach from a staminal tube (Figure 2C,D). Before its separation, this stamen is surrounded by unusually large intercellular spaces on its future lateral borders (Figure 2C). In *G. officinalis*, the morphology is quite similar. The vexillary stamen is extruded from a tube, although remaining connate with the adjacent stamens, but then detaches (Appendix A). The vexillary stamen is therefore morphologically somewhat distinct from the others. It cannot be excluded that this stamen unites with the adjacent ones at relatively late stages, resulting in a dedifferentiation of the contacting epidermal layers (this suggestion requires deeper investigation at earlier stages), but in the mature flower this stamen is not free, and there are no fenestrae at its base. We found it convincing to classify the androecium of *Galega* as monadelphous, which agrees with previous descriptions (e.g., [4]).

3.Pseudomonadelphous, with the vexillary stamen tightly attached to the adjacent adaxial stamens. When dissecting such androecia, it was required to apply a certain amount of force to detach this stamen from the androecial tube. This category was the most variable. Almost every examined genus (and sometimes even species) possessed a unique combination of features. The first source of variation involved the relative position of the vexillary stamen.3.1.Vexillary stamen superimposed: *Lathyrus* spp. (Figure 1C), *Trigonella foenum-graecum*, *Vicia hirsuta* (Appendix A).3.2.Vexillary stamen below adjacent stamens: *Melilotus officinalis* (Figure 1B).3.3.Vexillary stamen between two adjacent stamens: in this situation, the vexillary stamen can be either more or less terete in a cross section (*O. viciifolia*: Figure 1E, *V. sepium*: Appendix A), or flattened. In the latter case, the contacting margins of adjacent filaments are partly turned out, producing a kind of fin: *L. japonicus* subsp. *maritimus* (Figure 1F), *V. sylvatica* (Appendix A).

The second source of variation in pseudomonadelphous androecia was connected with the origin of the fenestrae at the base of the vexillary stamen.

3.a.Bases of the outer adaxial filaments are curved outwards, while the vexillary filament is straight: *O. viciifolia* (Figure 1E), *L. latifolius*, *V. hirsuta*, *V. sylvatica* (Appendix A).3.b.In addition to 3.a, the basal portion of the vexillary stamen is arched towards the adaxial side, producing a gibbosity at its base: *L. clymenum*, *L. niger*, *L. palustris* (Appendix A).3.c.In addition to 3.a, the basal portion of the vexillary stamen is bent towards the abaxial side: *V. sepium* (Appendix A) and possibly *Melilotus officinalis* (Figure 1B).3.d.No special curvature of the staminal bases occurs: *L. vernus* (Figure 1C), *L. japonicus* subsp. *maritimus* (Figure 1F). In this case, the fenestrae at the vexillary stamen base are very minute.

Patterns 3.1–3 and 3.a–d are associated with each other in different combinations. In *O. kamtschatica*, *T. foenum-graecum*, and *W. sinensis* (Appendix A), the curvature of the bases of the adaxial outer filaments is associated with a diadelphous androecium.

4.Diadelphous reduced, with inner stamens sterilized, i.e., substituted with antherless staminodes: *A. epiglottis* (Figure 1D), *A. pelecinus* (Appendix A). In both species, we were not able to examine the morphology of the vexillary stamen, but its filament (if any) seems to be free from the adjacent filaments, as there is a free margin along each adaxial outer stamen (Figure 1D; see also Appendix A).

It should also be noted that diverse modes of mutual arrangement of stamens can be observed on different levels of the androecium in some species. For example, in *M. officinalis,* the vexillary stamen has no contacts with the adjacent stamens in its lower and upper portions, while in its middle portion (less than half of its length) it is in tight contact with the adjacent stamens (Figure 1B). In other species, the mode of contact between stamens seems more or less uniform for the greater part of their length (Figure 1C,F,G).

### 2.2. Presence and Morphology of Floral Nectaries

Among species studied by us and listed in [4] as nectarless, only *O. spinosa* was confirmed as lacking any putative nectaries on the receptacle, hypanthium, inner side of staminal tube or carpel base (Figure 4K,L). All other species possessed modified stomata, which most probably act as nectaries. We possessed only herbarium material for *A. himalayanus*, and our observations yielded no reliable data regarding whether its flowers bear FNs or not. However, to our surprise, sparse, putatively nectar-producing stomata were detected in both species of *Astragalus* with sterile inner stamens, viz., *A. epiglottis* and *A. pelecinus* (Appendix A). We were unable to investigate the FNs of *Alhagi* in detail, as only herbarium material was available for examination, but there are numerous stomata on the receptacle and several on the hypanthium (Appendix A).

Stomatal pores in nectaries are sometimes clotted with a secrete and/or have granular deposits on their margins (Figure 3A). Morphology and sizes of these stomata differ from those of stomata on the calyx (Figure 3B,C). For example, in *M. albus,* the length of the stomatal pore on the inner calyx epidermis was 13.49 (11.01; 13.65) μm (here and below, data are given as median (1st quartile; 3rd quartile)) versus 10.99 (9.67; 11.31) μm for nectar-producing stomata (differences insignificant as per Mann–Whitney test). Similarly, in *A. cicer*, the length of the stomatal pore on the outer calyx was 13.25 (12.33; 14.86) μm versus 15.23 (14.13; 15.77) μm in the nectaries (differences insignificant as per Mann–Whitney test). Moreover, stomata in the nectaries and outside them may be of different types, e.g., actinocytic on the nectary of *A. cicer* (Figure 3B) and anisocytic on its calyx (Figure 3C), following the terminology reviewed in [20]. Sometimes, adjacent nectar-producing stomata were so close to each other that their guard cells contacted (Figure 3D).

To distinguish among different types of FNs, we focused on the position and abundance of nectariferous stomata, which differed among taxa. The areas of nectary secretion and release may not fully coincide; we used SEM-visualized external morphology as a key indicator. Compared with those of androecia, different morphologies of FNs are not easily separable. The following types were distinguished.

A high tube surrounding the base of a carpel with nectar-producing stomata only on the upper margin and inner surface, i.e., termed ‘automorphic’ type by [13]: *W. sinensis* (Figure 4A,B).A rim-like toroidal ridge surrounding a carpel’s base but with stomata present only on the outer abaxial side: *T. lupinaster* (Figure 4C,D), *T. medium* (Appendix A).An incomplete convex toroidal ridge around a carpel’s base. In this case, this ridge is lacking on the adaxial side, and nectar-secreting stomata are present only on its margin and (probably) the inner surface of the abaxial part: *L. clymenum*, *L. latifolius*, *L. niger*, *V. sepium*, *V. sylvatica* (Appendix A). In *V. sepium*, nectaries are borne on a ligulate abaxial outgrowth (Figure 4E,F). It is not easy to determine the exact position of nectaries of this type, but they are most probably placed on the hypanthium rather than on the receptacle, or just between them.An area bearing modified stomata without discernible elevation. Depending on the position of this area, it can be additionally classified into two subtypes.

Nectariferous stomata on the receptacle: *Galega* spp. (Figure 4I; see also Appendix A), *O. viciifolia*, *T. foenum-graecum*, *M. lupulina* (Appendix A).

4.1.Nectariferous stomata on the hypanthium: *Astragalus* spp. (Figure 4G), Ca. *arborescens*, *L. japonicus*, *L. palustris*, *M. officinalis*, *O. kamtschatica* (Appendix A) Again, it is not easy to unambiguously decide if nectaries are on the receptacle or the hypanthium in *Lathyrus*.4.2.When nectaries are located in an area without borders, this area usually has an abaxial position. Only in *Melilotus* does this area seem to be expanded to the whole basal circumference of the hypanthium, or most of its surface.

In addition to stomata on the receptacle and hypanthium, we observed stomata on the connectives of stamens (Appendix A), but not on filaments, in some of the examined species (*Co. arborescens*, *G. officinalis*, *O. kamtschatica*, *V. sepium*, *W. sinensis*). Not all species were examined for this feature, and not all available material provided the possibility of such analysis. Nectaries or other secretory structures on the outer (abaxial) surfaces of anthers were previously reported in Leguminosae ([21] and papers cited therein), as well as in other angiosperm families, both basal and derived [22,23]. To test whether these stomata are of a secretory nature in the IRLC, deeper examination is required, applying both histochemical tests and observations of living plants.

### 2.3. Corolla Abnormalities Associated with Monadelphy in Astragalus albispinus

The anomalous specimen of *A. albispinus* from the TARI herbarium (see Materials and Methods) first attracted our attention with its monadelphous androecium. However, a closer examination revealed that its flowers also possess anomalous petals. Normally, this species is characterized by a standard petal not differentiated into limb and claw, which is characteristic of the entire *Adiaspastus* section to which it belongs [24]. Only a shallow constriction delimits the border between the limb and claw of a flag, while keel and wing petals are clearly unguiculate (Figure 5A). However, in the flowers of the anomalous specimen, a flag also displays a narrow unguis (Figure 5B,C). The remaining petals seem differentiated properly. For example, in both normal and abnormal specimens, lateral petals (wings) have a specific ridge-like sculpture on the outer surfaces of their limbs (Figure 5F,G).

The epidermis of atypical flags is composed of smaller cells than that of normal flags (Figure 5D,E). The measurements of 20 cells of the outer (adaxial) epidermis yielded results of 409.8 ± 124.2 μm^2^ and 662.7 ± 168.1 μm^2^ for atypical and typical flags, respectively (average ± standard deviation; differences significant as per Mann–Whitney U test, *p* < 0.01).

## 3. Discussion

### 3.1. Evolutionary Trends of Androecia and Floral Nectaries in the IRLC

When comparing the results of our SEM studies on FNs in the IRLC with reliable published data (especially accompanied with illustrations), we might conclude that most of the representatives of this group possess FNs of a very similar structure. Their secretory areas represent modified nectar-producing stomata on the abaxial part of the receptacle between the carpel and the stamens (Figure 4). In some cases (*Astragalus*), such stomata are also found on the abaxial side of the hypanthium (Figure 4G). Among studied genera, only *Wisteria* has a circular pipe-like nectary surrounding the carpel base that bears stomata on its inner surface (Figure 4A). The latter morphology seems more typical of the Phaseoleae tribe and allied groups [25]. Among robinioids, which form a sister group with the IRLC [2], one may find nectaries either localized abaxially on the receptacle and hypanthium (*Lotus*: [26]) or distributed on all surfaces of the hypanthium’s lower portion (*Robinia*: [27] and papers cited therein). In several genera (*Trifolium*, *Vicia*), FNs had a somewhat intermediate structure with complete toroidal elevation around a carpel base, but with secretory stomata present only on the abaxial side (Figure 4C; see also Appendix A).

To date, nectaries have been characterized in only a few genera in the NPAAA clade (this includes phaseoloid, mirbelioid, and robinioid genera, the IRLC, and some other groups: [2]). However, we may preliminarily conclude that the main trend of nectary evolution is towards size reduction, together with a tendency towards a preferentially abaxial position. Ancestral concave annular nectaries, either free (*Wisteria*) or adnate to the hypanthium base, are placed at the abaxial side in most of the IRLC, sometimes still elevated above the receptacle surface (*Lathyrus*, *Vicia*) but more often not. We agree with the conclusion of Gulyás and Kincsek [13] that monosymmetric nectaries are the most specialized, but not with their hypothesis that the absence of nectaries is the ancestral state. Nectaries are lost repeatedly and independently in several papilionoid lineages, such as the Genisteae tribe or *Ononis* within the Trifolieae.

Surprisingly enough, nectaries in the IRLC are very ‘inertial’ in the evolutionary sense. They persist, although small in size, in taxa with very miniaturized flowers, such as *M. lupulina* (Appendix A), although Gupta [28] reported that a certain fraction of flowers of *M. lupulina* and some other *Medicago* species might have no nectariferous disc. *M. lupulina* was found to be not only self-compatible but also possessing cleistogamous flowers with pollen germinating in anthers [29]. The existence of nectar-producing stomata in tiny flowers of this medic agrees with observations of insects visiting its flowers and providing occasional cross-pollination (reviewed in [30]). Similarly, nectaries are found in the flowers of small-flowered vetches, such as *V. hirsuta* and *V. tetrasperma* [15]. Insects may be required not only as agents of pollen transfer but also to promote the so-called ‘insect induced self-pollination’ (e.g., [31]), so there may be a need to attract insects even for self-pollinating flowers.

Reduction of the inner staminal whorl, as in *A. epiglottis* and *A. pelecinus*, which hypothetically implicates a trend towards preferential self-pollination [32], also does not completely remove the FNs from flowers of these species (Appendix A).

Our data are in agreement with previous reports that flowers of *Ononis* are nectarless [4]. However, both examined species of *Galega*, although possessing monadelphous androecia, bear receptacular nectar-producing stomata (Figure 4I,J; see also Appendix A). Our observations on the anomalous specimen of *A. albispinus* indicate that monadelphy itself does not prevent the development of FNs; therefore, these two ontogenetic programs are separable. Indeed, FNs may be found in papilionoid flowers with free stamens (*Anagyris*: [4]; *Thermopsis*: [33] and papers cited therein), di- (numerous genera) or monadelphous (*Galega*, anomalous *A. albispinus*: see text) androecia, or with a pseudomonadelphous androecium. This suggests that staminal fusion, at least of certain types, does not immediately influence the subsequent evolution of FNs. However, if FNs were to disappear, this would cause a shift in reproductive strategy of a species. For example, nectarless leguminous flowers evolved heteromorphic stamens, with some of them producing pollen as a reward for the pollinator ([34] and papers cited therein). Although there are not many data concerning the association between the morphology of nectaries and staminal fusion, there is probably only one reported case of a nectarless papilionoid flower with a diadelphous androecium, viz., *Securigera varia* (L.) Lassen (tribe Loteae: [35]), although this case requires special revisiting. Most IRLC species reported as having diadelphous nectarless flowers by Rodríguez-Riaño et al. [4] were marked by them as potentially necteriferous, and deserve reexamination via SEM.

The results reported here, together with data previously acquired for other species of the same tribe [14,15,16,36,37], drive us to the conclusion that androecial and FN features may both be relatively constant, at least within each genus. However, both androecia and FNs may be morphologically diverse in some genera (see below).

Different androecium morphologies may be adapted towards different modes of interaction with pollinators. When there are wide gaps between the free stamen and adjacent stamens (such as in *Astragalus* spp., *Caragana*: Figure 1A), insects may access the nectar-accumulating chamber between the carpel base and the staminal tube.

The collection of nectar from a specialized nectar chamber also seems to be possible in pseudomonadelphous androecia with large fenestrae at the vexillar stamen base, sometimes additionally widened due to curvature of its filament, as well as curvature of filaments of adjacent stamens (Figure 1B,E). Westerkamp [38] reported that access is usually only available from one side in the asymmetric flowers of *L. latifolius*. When fenestrae are very small (Figure 1C,F), insects may collect nectar outflowing to the space between the staminal tube and the perianth. Finally, in flowers with a monadelphous androecium, nectar may accumulate inside the staminal tube and be available for pollinators there. A completely fused staminal tube is associated with abundant nectar secretion in some caesalpinioid legumes, such as *Inga* [39] and *Enterolobium*, but is not a common syndrome in legumes.

### 3.2. Androecial and Nectary Features Have Low Taxonomic Value in the IRLC

It is difficult to define any common feature of androecium morphology in the IRLC. A common ancestor of this group most probably possessed a diadelphous androecium, which then repeatedly gave rise to either monadelphous or pseudomonadelphous types. Some traditionally recognized tribes unite members with different androecial morphologies (such as di- and pseudomonadelphous in the Fabeae, mon- and pseudomonadelphous in the Trifolieae etc.). Going deeper into the details complicates this situation further. For example, in *Melilotus* (Trifolieae) a flattened filament of the vexillary stamen goes beneath the other stamens (Figure 1B), while in *Trigonella* (Appendix A) and *Medicago* (Figure 12 in [14]) from the same tribe, this filament is clearly superimposed. A superimposition of the vexillary stamen is also found in several other genera (*Lathyrus*, *Vicia*), so different habits seem to arise repeatedly and independently in different lineages, hence reducing the value of this feature for systematics.

Similarly, a gross morphology of FNs is relatively conserved throughout the IRLC. Among examined genera, only *Wisteria* has a distinct localization of nectaries (Figure 4A), while other genera possess more or less abaxialized secretory stomata on a more or less elevated area. However, even within a comparatively diverse genus such as *Lathyrus,* one may find variable habits of FNs.

### 3.3. Staminal Fusion Is Related to Corolla Morphology

As seen from the case with the anomalous flowers of *A. albispinus*, the androecium becomes monadelphous together with the deformation of a flag, i.e., these two abnormalities arise as a complex syndrome. A similar situation was previously reported in unusually polysymmetric flowers of *Etaballia* (Dalbergieae tribe), where a completely closed staminal tube was associated with atypical ribbon-shaped petals, most probably resulting from deficiency in marginal growth [40]. The latter morphology is not common among allied genera, having a monosymmetric flag blossom with unguiculate petals. Conversely, in the case of anomalous dorsalization of the corolla, when all five petals become flag-like, all inner stamens become free from fusion [41].

The floral monosymmetry in most angiosperms is principally regulated by the TCP genes (such as the well-known *CYCLOIDEA* discovered in *Antirrhinum majus* L. of the Plantaginaceae: [42]). While this regulation is usually discussed with respect to corolla (mono)symmetry, patterns of staminal differentiation are also involved. As demonstrated by Hsu et al. [43], the expression of the *CYC* orthologue defines both the specialization of two adaxial petals and the conversion of three adaxial stamens into staminodes in *Saintpaulia ionantha* H.Wendl. (Gesneriaceae). When this gene’s expression is expanded into all petals, all stamens become staminodial, while reduction or loss of adaxial expression leads to fertilization of all stamens. The expression of *CYC*-like gene was demonstrated not only in a future flag but also in the primordium of the vexillary stamen in *Lotus* [44]. The ectopic expression of *CYC*-like gene in all five petals in this model plant resulted in completely free stamens. These results, together with the observations reported here, indicate that corolla monosymmetry and features of the androecium are tightly connected in developmental regulation. It is therefore not surprising that the flowers of *A. albispinus* with completely fused stamens also have an abnormal flag shape (Figure 5B,C).

However, alterations in corolla symmetry are not a unique source of androecium evolution. As seen from the examples of *Ononis* or *Galega*, stamens may fuse without changes in perianth monosymmetry. The simplest way to reach such a state is probably to shift the expression pattern of *CYC*-like genes so that their activity would persist in the adaxial petal (thus defining its conversion into a flag) but be absent from the adaxial stamen (and this stamen would no longer have any special features). To test this, detailed examination of the spatial and temporal features of *CYC*-like gene expression are required in leguminous flowers with different androecial morphologies.

## 4. Materials and Methods

### 4.1. Plant Material

Plant materials used for study are listed in Table 1. In total, 29 species of 15 genera were sampled for this work. Together with an additional 23 genera of the IRLC, all of these were included in molecular phylogenetic work by [2]. Our genera represent most of the subclades of the IRLC revealed by [2].

Freshly collected flowers were fixed in 70% ethanol, dissected under a stereomicroscope and prepared for electron microscopy as described below. For several species, only herbarium specimens were available. These species (such as *A. epiglottis*, *A. himalayanus*, and *A. pelecinus*) were chosen mostly because of their previously reported unusual androecial morphology [4,32], so we decided to include them in our investigation despite the unavailability of fresh specimens. Desiccated flowers were soaked in hot water (90–95 °C) and then placed in 70% ethanol and thermostated at 60 °C for 24 h. After this treatment, the material was dissected and prepared for SEM.

During study of *Astragalus* species in Iran, we encountered an unusual specimen of *Astragalus albispinus* Širj. & Bornm., an endemic species with an anomaly in the staminal tube. A large number of herbarium specimens of *A. albispinus* and other related taxa from *Astragalus* sect. *Adiaspastus* were examined in detail in terms of flower structure. However, one of authors (A.B.) undertook field expeditions during the spring/summer of 2020 to the vicinity of Borujen (the locality from which the abnormal specimen was collected) and other adjacent areas in Chaharmahal and Bakhtiari province, and was not able to observe any abnormality in flowers of this taxon in nature. It is therefore unclear whether this unusual habit was a heritable feature or resulted from some kind of environmental influence. We focused on studying the only abnormal herbarium specimen of this species, with the following collection specifications: Iran, Chaharmahal and Bakhtiari, Borujen, ca. 9 km from Naghneh to Tang-e Ahan, 2400 m.a.s.l., 24 July 1998, Maassoumi and Mozaffarian 76739 (TARI).

In addition to morphological traits, we also decided to use molecular data to accurately identify the abnormal specimen. For this purpose, both the normal and abnormal specimens of *A. albispinus* (TARI57878 and TARI76739, respectively) were used for molecular phylogenetic analysis of the nuclear ribosomal DNA internal transcribed spacer (ITS) region as well as the plastid *ycf1* and *matK* genes. Both samples were found to have identical sequences, so the abnormal specimen certainly belongs to *A. albispinus*. For more detailed information regarding the molecular phylogenetic analysis of these samples and their GenBank (NCBI, USA) accession numbers, see [45].

### 4.2. Scanning Electron Microscopy (SEM)

For SEM studies, floral buds were fixed and stored in 70% ethanol, then dissected under a stereomicroscope and dehydrated for SEM with ethanol (80% and 96% for 60 min each), a mixture of 96% ethanol:acetone (1:1, 60 min), and a final dehydration in acetone (60 min). Material was subsequently dried using a HCP-2 (Hitachi, Tokyo, Japan) critical point dryer, mounted onto metal stubs using nail polish, and coated with Pd in an Eiko IB-3 (Eiko, Tokyo, Japan) sputter coater. SEM images were taken with a CamScan-S2 (Cambridge University, Cambridge, UK) microscope in a Secondary Electron Image (SEI) regime with an acceleration voltage of 20 kV.

### 4.3. Anatomy

For anatomical studies, ethanol-fixed floral buds were dehydrated via ethanol series (80% ethanol for 45 min, 96% ethanol for 60 min twice, 100% ethanol for 60 min twice), a mixture of 100% ethanol:chloroform (3:1, 1:1, 1:3), and a final dual dehydration in pure chloroform (in each of these stages, material was stored until it submerged into liquid). Then, the material was saturated with paraffin at 56 °C for a week [46]. Transverse sections of 15 μm in thickness were obtained with a rotary microtome HM355S (Thermo Scientific, Waltham, MA, USA), mounted on slides, washed with xylol (twice for 30 min) and 96% ethanol (twice for 30 min) to remove the paraffin, rinsed with distilled water and stained with a 0.5% water solution of toluidine blue for 5 min. Slides were then rinsed with distilled water, dehydrated via ethanol series (70%, 96%, 100% for 60 min) and xylol [46] or Bio-Clear (Bio-Optica Milano S.p.A., Milano, Italy), and embedded in VitroGel medium (TheWell Bioscience, North Brunswick, NJ, USA). Sections were visualized and photographed with an AxioPlan 2 (Zeiss, Germany) microscope equipped with a Zeiss AxioCam MRc 5 (Zeiss, Oberkochen, Germany) digital camera.

Some images were captured with an Olympus SZ61 stereomicroscope (Olympus, Tokyo, Japan) using a UHCCD05000KPA camera (ToupTek Photonics, Hangzhou, China). All measurements on digital images were carried out using the program ImageJ 1.51k (National Institutes of Health, USA) and statistically treated with Statistica 12 (Statsoft, Tulsa, OK, USA).

Digital images were prepared for publication via processing with Corel PHOTO-PAINT 2017 (Corel Corporation, Ottawa, ON, Canada). Only representative morphologies are illustrated with figures in the text, while images of the androecia and nectaries of all other genera are given in Appendix A.

## 5. Conclusions

Despite the general similarity of floral (especially corolla) morphology across members of the IRLC, their androecia are very diverse regarding vexillary stamen connation and position with respect to adjacent stamens, as well as shape and size of fenestrae. Different variations may coexist in the same genera. Conversely, features such as the superposition of the free stamen or its congenital fusion with other stamens, producing a monadelphous androecium, evolved repeatedly in different lineages. The staminodial conversion of inner stamens occurred in *Astragalus* at least twice in different sections. All these events suggest that different androecial morphologies are of little taxonomic importance due to a high level of homoplasy, although they are undoubtedly intriguing for studies of adaptations towards different pollination strategies.

As for FNs, they also have a common gross morphology in most of the examined representatives of the IRLC, i.e., they represent an area of secretory stomata on the receptacle and/or hypanthium, preferentially on the abaxial side. Although their sizes and distribution may vary even within large genera (*Lathyrus*, *Vicia*), FNs in the IRLC seem more conserved than androecia. This conclusion may partly result from the fact that it is much more difficult to distinguish between different types of FNs in this group, i.e., their habits are much less discrete than androecial types. It may be concluded that FNs are considerably conserved. Surprisingly enough, FNs are associated even with floral miniaturization, reduction of the inner staminal whorl or, in the case of *Galega*, the switch to monadelphy. All these processes are believed to assist preferential self-pollination, which, in particular cases, still requires visitation by insects. As evidenced from the cases of *Galega* and the anomalous monadelphous form of *Astragalus*, staminal fusion and the presence of FNs are separable features on both regulatory and evolutionary levels.

## Figures and Tables

**Figure 2 plants-11-00649-f002:**
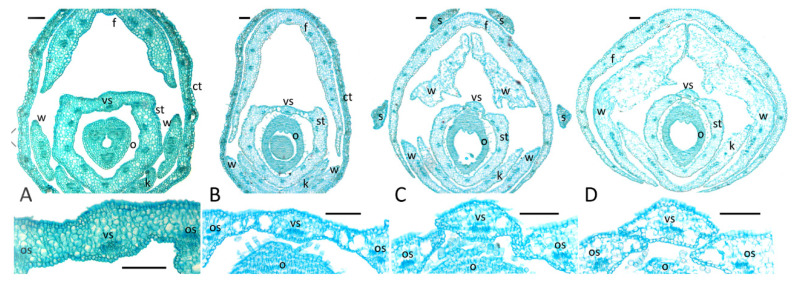
Androecium fusion in *Galega orientalis* as seen in a series of transverse sections of the preanthetic flower on different levels ((**A**) is closest to the receptacle, while (**D**) is the most distal position) via light microscopy. On each level, the upper image represents overall topography, while the lower is a magnification of the vexillary stamen. All images are oriented with their adaxial sides upwards. Key: ct = calyx tube; f = flag; k = keel; o = ovary; os = outer adaxial stamen (as traced by its vascular bundle); s = sepal; st = staminal tube; vs = vexillary stamen (**C**,**D**) or its bundle (**A**,**B**); w = wing. Scale bars: 100 μm.

**Figure 3 plants-11-00649-f003:**
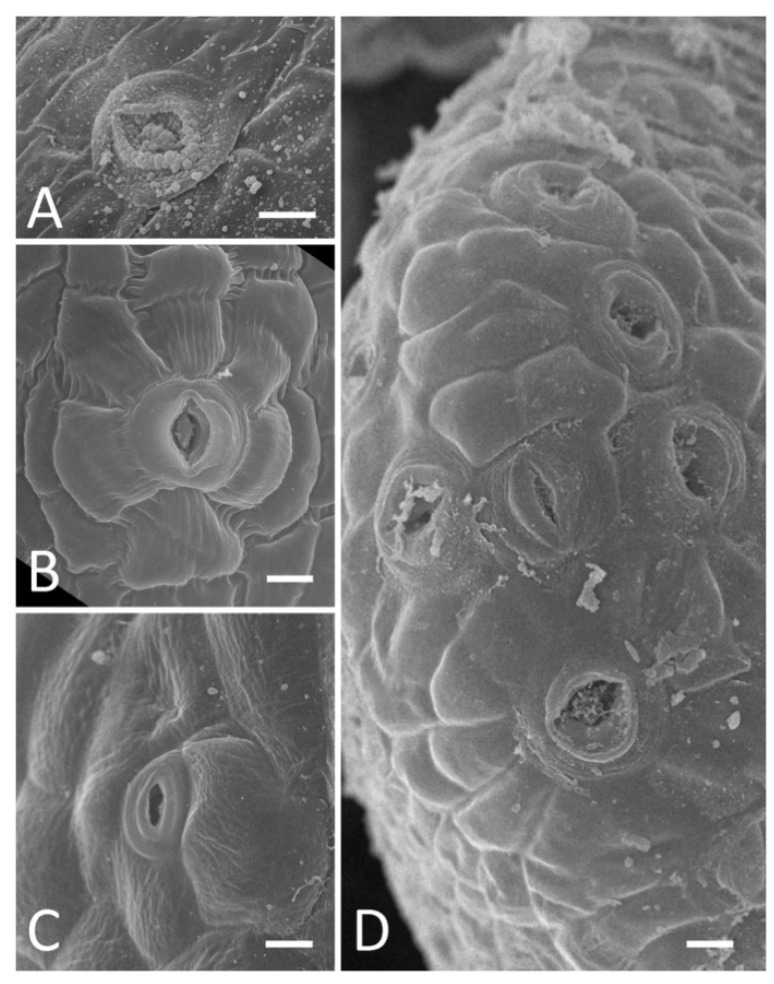
SEM images of stomata on nectaries (**A**,**B**,**D**) and calyx (**C**). (**A**), *Vicia sylvatica*; (**B**,**C**), *Astragalus cicer*; (**D**), *V. sepium*. Images (**B**–**D**) have the same magnification. Scale bars: 10 μm.

**Figure 4 plants-11-00649-f004:**
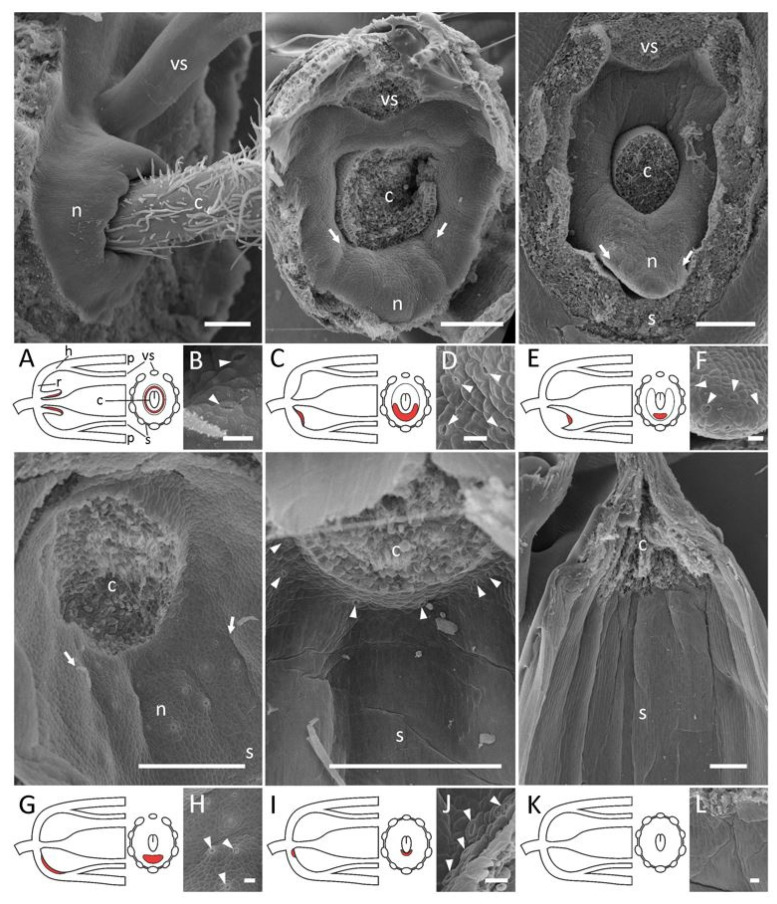
Localization of floral nectaries in selected representatives of the IRLC. (**A**,**C**,**E**,**G**,**I**) represent the overall morphology (above, SEM) and schematic representations of nectaries in side (below left) and top (below right) views. (**K**) depicts a nectarless floral cup (SEM). (**B**,**D**,**F**,**G**,**J**,**L**) are closer views of nectariferous areas, or a similar region in nectarless flower (**L**). All or most floral organs are removed. (**A**,**B**), *Wisteria sinensis*; (**C**,**D**), *Trifolium lupinaster*; (**E**,**F**), *Vicia sepium*; (**G**,**H**), *Astragalus cicer*; (**I**,**J**), *Galega orientalis*; (**K**,**L**), *Ononis spinosa*. Key: red color = secretory area (on schemes); arrows = outer borders of secretory area (on SEM images); arrowheads = exemplary nectariferous stomata; c = carpel or place where it was localized; h = hypanthium; n = nectariferous area; p = perianth; r = receptacle; s = staminal tube; vs. = vexillary stamen or place where it was localized. Scale bars: 300 μm (**A**,**C**,**E**,**G**,**I**,**K**), 30 μm (**B**,**D**,**F**,**H**,**J**,**L**).

**Figure 5 plants-11-00649-f005:**
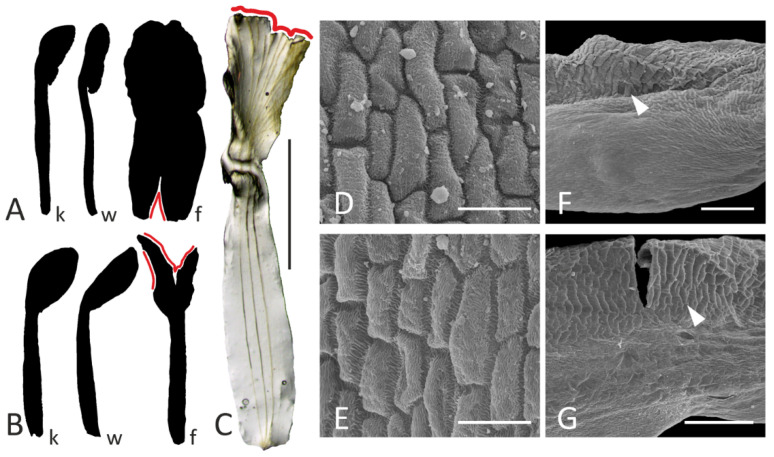
Corolla morphology in normal (**A**,**D**,**F**) and abnormal (**B**,**C**,**E**,**G**) flowers of *Astragalus albispinus*. (**A**,**B**): contours of petals; (**C**): flag of abnormal flower (stereomicroscope image) with venation visible; (**D**–**G**): epidermal cells (SEM image) of flag (**D**,**E**) and wing (**F**,**G**) external surfaces. Key: f = flag; k = keel petal; w = wing; red line = portion of petal damaged in the course of dissection; arrowhead = ridge-like sculpture. Scale bars: 2 mm (**C**), 300 μm (**F**,**G**), 30 μm (**D**,**E**).

**Table 1 plants-11-00649-t001:** Plant material used for the study.

Species	Origin of Material	Voucher Accession
*Caragana arborescens* Lam.	Russia, Moscow region, ornamental	No voucher
*Vicia hirsuta* (L.) Gray	Russia, Moscow region	MW0568648
*V. sepium* L.	Russia, Moscow region	MW0568642
*V. sylvatica* L.	Russia, Moscow region	MW1064058
*Lathyrus clymenum* L.	Origin unknown, reproduced from seeds	MW1064054
*L. japonicus* subsp. *maritimus* (L.) P.W.Ball	Russia, Kamchatka	MW0165477
*L. latifolius* L.	Russia, Moscow region, ornamental	No voucher
*L. niger* (L.) Bernh.	Russia, Kaluga	No voucher
*L. palustris* L.	Russia, Kamchatka	No voucher
*L. vernus* (L.) Bernh.	Russia, Moscow region	MW0568640
*Astragalus cicer* L.	Russia, Moscow	MW0568650
*A. albispinus*	Iran	TARI76739 (abnormal), TARI57878 (normal)
*A. caspicus* M.Bieb.	Iran	TARI54008
*A. himalayanus* Klotzsch	India	MW0740469, MW0740470
*A. epiglottis*	Morocco	MHA Blanché et el. 9785
*A. pelecinus*	Portugal	MHA, Matos et al. 6634
*Galega orientalis* Lam.	Russia, Moscow	MW1066283
*G. officinalis* L.	Russia, living collection of the ‘Aptekarskiy Ogorod’ botanical garden	No voucher
*Medicago lupulina* L.	Russia, Moscow	MW1072490
*Melilotus officinalis* (L.) Pall.	Russia, Moscow	MW1072489
*Ononis spinosa* L.	Russia, Kaluga region	MW1066275
*Trifolium medium* L.	Russia, Moscow region	MW1072488
*T. lupinaster* L.	Russia, Murmansk region	MW0408297
*Trigonella foenum-graecum* L.	Origin unknown, reproduced from commercially available seeds	MW1066273
*Oxytropis kamtschatica* Hulten	Russia, Kamchatka	MW0954585
*Onobrychis viciifolia* Scop.	Russia, living collection of the Lomonosov Moscow State University botanical garden	No voucher
*Wisteria sinensis* (Sims) Sweet	Russia, living collection of the Tsitsin Main Botanical Garden	No voucher
*Colutea arborescens* L.	Russia, living collection of the Lomonosov Moscow State University botanical garden	No voucher
*Alhagi maurorum* Medik.	Russia, Astrakhan	MW0416369

## Data Availability

Not applicable.

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
