# Peer review of "Variations in Structure among Androecia and Floral Nectaries in the Inverted Repeat-Lacking Clade (Leguminosae: Papilionoideae)"

_plants, 2022, doi:10.3390/plants11050649_

Round 1
Reviewer 1 Report
The paper by Sinjushin et al. is dedicated to patterns of androecial construction and nectary location in a diverse and species-rich clade of Leguminosae. The study itself is very interesting. I have some question and concerns that would help to improve the manuscript.
What was the reason for taxon sampling? The authors examined about 30 species. Some specimens were taken from herbaria. Why? Herbarium material is not the best one for SEM and anatomical studies. I wonder to what extent is sampling comprehensive with respect to the phylogeny?
It seems that authors fail to interpret their results unequivocally in some places. For example, lines 106-108: ‘Diadelphous…. Caragana arborescens (Fig. 1A), Astragalus albispinus (normal), A. caspicus, A. cicer, Colutea 107 arborescens, Oxytropis kamtschatica, and PROBABLY Wisteria sinensis’.
My main concerns are regarding monadelphous vs pseudomonadelphous androecia. The best way to distinguish between these patterns is to study floral development either under SEM or anatomically. The interpretation of androecium of Galega as monadelphous is not convincing. Indeed, there is an area at the base of the staminal tube where all the stamens are apparently congenitally united. But the absence of lines of postgenital fusion (contacting epidermises) does not automatically mean monadelphy. In many cases, postgenital fusion results in deep re-differentiation of contacting epidermises without leaving any traces of initial organ borders (see classical works of Verbeke).
The same is appliable to pseudomonadelphous vs diadelphous androecia. As I understand, the vexillary stamen should be postgenitally fused to adjacent stamens in pseudomonadelphous androecia. Line 132 contains somewhat controversial statement ‘Pseudomonadelphous, with the vexillary stamen contacting with adjacent adaxial stamens.’ ‘Contacting’ is confusing here. Note, that tight contact should be distinguished from postgenital fusion (see Verbeke and Endress). The authors provided only SEM photos for all cases of pseudomonadelphous androecia. To prove pseudomonadelphy, an anatomical examination is needed (late floral buds are most helpful).
Authors presented three nectary types where there is rim around the gynoecium base. These types differ by nectary location on this rim (fig 4 a-c). How do you distinguish between those types? Only by nectary stomata location? Note, that areas of nectary secretion and nectary release do not always have the same location. Again, anatomical sections would help much to recognize nectary secreting areas.
For other comments see attached pdf.
Author Response
Dear colleague,
We are very grateful for your work on reviewing of our manuscript. According to your suggestions, we have made numerous changes, which you may find in the attachment (for your convenience, all changes were marked with red). Here are the explanations and descriptions of these updates, as well as responses to your questions.
1. We sampled 15 genera which represent most of molecularly-recognized subclades within the IRLC. The explanation of this is now added to the text (lines 458-461). As for herbaria, we attempted to investigate mostly those species, which were previously described as having unusual, remarkable androecial morphology (such as some of Astragalus). Indeed, at least two of herbarium-preserved species were not amenable for SEM analysis, but we left their names in the Materials and Methods section.
2. Indeed, in some cases our protocols brought doubtful. Fortunately, such situations were not often, so in most species we managed to judge about their androecial and nectaries' morphology unambigously.
3. The controversy of adelphy types can be reduced to the problem of what we define as fusion. We expanded this part in the Introduction (lines 79-94) and added two references, including a paper by Verneke according to your suggestion. The review of literature indeed proves that 'perfect postgenital' fusion is indistinguishable from congenital fusion, if judge by anatomy of mature flowers. Therefore we have retained our idea that androecium of Galega can be classified and monadelphous (which agrees with previously reported descriptions), although this case really deserves a special survey in ontogenetic dynamics. Hopefully we will manage to do it in a future, it is an interesting challenge.
4. For the same reasons, we interpreted pseudomonadelphy broadly enough, including all cases when the vexillary stamen was separable from adjacent stamens but applying some force, evidencing for some kind of adhesion. However, to avoid confusion, we rephrased descriptions of androecial types, sometimes with direct quotations (lines 61-63, 90-93).
5. We agree that areas of nectar biosynthesis and its secretion may be different. Some of our statements were rephrased accordingly.
6. As for Fig. 4, we find three first types distinguishable by the location of secretory area and presence/absence of rim-like elevation. Fig. 4E was adjusted to make this difference more evident.
7. We have made numerous rewordings and language corrections according to your comments directly in the file.
We thank you again for your helpful comments, as they were really challenging and promoted us to make a text more accurate in its statements. We hope that in its updated version our manuscript can be recommended for publication in Plants.
Many thanks and best regards,
Authors.

Reviewer 2 Report
The manuscript represents interesting study on the morphology of androecium, in particular position of vexillary anther, and floral nectaries (position of stomata and access to nectar) in IRLC ( Inverted Repeat Loss Clade). The science is good quality and my general opinion on the manuscript is very positive, but some items need attention of the authors. Namely: (line 71) change to “sterile”. Also, I suggest to delete from Results the part on the stomata present on the conectives (2.4. Putative Secretory Stomata on Staminal Connectives ) because it does not bring any information and is speculative. The same, and from the same reason, should be deleted from Discussion (3.4). With reference to M&M, I encourage to use standard procedures for SEM (fixation in glutaraldehyde, not in 70% etOH which can cause shrinkage/collapse of the cells. Also in paraffin method, after deparaffinization in xylene, important step is to hydrate slides before staining in TBO. Please check and provide reference for your lab protocol. My main reservation concern illustrations, in particular Fig 4. I recommend to keep the same position of scale bars and all letters on the figures throughout the manuscript (e.g right bottom corner for the scale bar and left upper corner for letter). I also suggest to rearrange illustrations in such manner that vexillary stamen is always in the same orientation. This will introduce some order and will be very helpful for readers. I do not understand why large SEM images in Fig. 4 are without letters. Large SEM representing Galega orientalis (?) is bad quality and upper part (above “c”) should be cut. I also recommend revision of SEMs in S1. Some images are bad quality, e.g. Alhagi maurorum, Astragalus epiglottis, Astragalus pelecinus, Galega officinalis, L. palustris, they have no letters (numbers) and the caption to the figures is insufficient.
Author Response
Dear colleague,
We are very grateful for your work on reviewing of our manuscript. According to your suggestions, we have made numerous changes, which you may find in the attachment (for your convenience, all changes were marked with red). Here are the explanations and descriptions of these updates, as well as responses to your questions.
- We have substantially expanded the Materials and Methods section and described all procedures, especially those connected with microscopy (lines 506-516). Indeed, there is a stage of rehydration of slides before staining, now it is clear from the text.
- Thank you for your recommendations concerning material fixation and further processing. We have tested different ways of fixation (such as ethanol or glutaraldehyde) in the course of our previous studies. Our experience evidences that fixation of plant material in 70% ethanol does not deny the opportunity of further SEM studies even in the case of floral ontogeny, when we examine meristems. Stamens and nectaries (as well as other mature floral structures) are usually tough enough, if fixed freshly collected.
- As for figures (e.g., Fig. 4), we have rearranged them to make positions of letters and scale bars constant wherever possible. In several cases (Fig. 1G, H; 6D), we have left the initial design, as it is difficult to reformat them due to the character of image itself (letter/scale bar would hide some important parts).
- In Fig. 4I, we do not possess image(s) of a better quality. The upper part of this SEM photo can be deleted but we retained it for a sake of the whole figure's symmetry. This part, although of imperfect quality, does not interfere with reception of this photo.
- Large images in Fig. 4 have the same lettering as the corresponding schemes. That is why we used letter designations for SEM photos and line schemes simultaneouly. We hope it will be clear for readers from the corresponging caption.
- In Supplement 1, we added more designations on photos and adjusted captions accordingly. In our idea, there is no need to designate images in Supplement 1 with letters/figures, as each image has its own caption.
We thank you again for your helpful comments, which indeed made our text more readable, as well as for positive evalution of this work. We hope that in its updated form this paper can be recommended for publication in Plants.
Many thanks and best regards,
Authors

Round 2
Reviewer 1 Report
The authors have improved the ms and clarified all the point raised in my review. I would suggest adding one more reference J. A. Verbeke 1992 FUSION EVENTS DURING FLORAL MORPHOGENESIS https://www.annualreviews.org/doi/pdf/10.1146/annurev.pp.43.060192.003055?casa_token=a1kFxP-4BakAAAAA:b38VQdtWMiJwhPOQe6L63eF-XmVhHezJ-vgF2993i1As8n5p_bzoy8M1dm7CI4v6RpU2gOO3V0RJK-E
Author Response
Dear colleague,
Thank you for your helpful suggestions and positive evaluation of our work. We have added the recommended reference and renumbered citations accordingly (see attached; for your convenience, all changes are marked with red).
Best regards,
Authors

Reviewer 2 Report
Dear Author
I reviewed the new version of the manuscript, and I noted that almost all my comments were incorporated. Unfortunately, authors did not refer to my suggestion related to removing from the manuscript (Results and Discussion) all parts on staminal connectives as additional secretory sites. I uphold my opinion that this paragraph is too speculative and do not bring any valid information. Also, some marks on illustrations, namely letters and scale bars are not uniformly distributed, but this is only technical problem.
Author Response
Dear colleague,
Thank you for your helpful suggestions and positive evaluation of our work.
We don't feel that our observations of stomata on staminal connectives are completely meaningless. They provide a clue to the future researches, as this ambiguity definitely deserves a deeper examination. However, we agree that at the moment any hypotheses about function of these stomata are somewhat unfounded. We have reduced the discussed part of the text till a single paragraph in the Results section (not a separate subsection, as it was before), see lines 269-277 in the attached file (for your convenience, all changes are marked with red).
We have completely removed a corresponding subsection from the Discussion. A former Fig. 6 with its caption is no longer present in a text but moved to the new Supplement 3.
As for scale bars and letters on illustrations, as mentioned before, we made their positions as uniform as possible without damaging images and hiding important details. We hope that this flaw (if consider it like this) would not interfere with perception of the whole paper.
We thank you again and hope that in its present form our paper can be considered acceptable for publication in Plants.
Best regards,
Authors
